# AN EVOLUTIONARY PERSPECTIVE ON MODES OF LEARNING IN TRANSFORMERS

**Alexander Y. Ku**[1,2]**, Thomas L. Griffiths**[2]**, Stephanie C.Y. Chan**[1]
[1]Google DeepMind
[2]Department of Psychology, Princeton University
`{alexku, scychan}@google.com, tomg@princeton.edu`

## ABSTRACT

The success of Transformers lies in their ability to improve inference through two complementary strategies: the permanent refinement of model parameters via *in-weight learning* (IWL), and the ephemeral modulation of inferences via *in-context learning* (ICL), which leverages contextual information maintained in the model's activations. Evolutionary biology tells us that the predictability of the environment across timescales predicts the extent to which analogous strategies should be preferred. Genetic *evolution* adapts to stable environmental features by gradually modifying the genotype over generations. Conversely, environmental volatility favors *plasticity*, which enables a single genotype to express different traits within a lifetime, provided there are reliable cues to guide the adaptation. We operationalize these dimensions (environmental stability and cue reliability) in controlled task settings (sinusoid regression and Omniglot classification) to characterize their influence on learning in Transformers. We find that stable environments favor IWL, often exhibiting a sharp transition when conditions are static. Conversely, reliable cues favor ICL, particularly when the environment is volatile. Furthermore, an analysis of learning dynamics reveals task-dependent transitions between strategies (ICL $\rightarrow$ IWL and vice versa). We demonstrate that these transitions are governed by (1) the asymptotic optimality of the strategy with respect to the environment, and (2) the optimization cost of acquiring that strategy, which depends on the task structure and the learner's inductive bias.

## 1 INTRODUCTION

Transformers (Vaswani et al., 2017) underpin the success of modern large language models (LLMs), demonstrating impressive performance across a diverse set of tasks (Brown et al., 2020). A key emergent capability of these models is *in-context learning* (ICL), which allows them to perform novel tasks specified by examples in the input prompt, without updating model parameters (Brown et al., 2020; Dong et al., 2022; Liu et al., 2023). This stands in sharp contrast to standard *in-weights learning* (IWL), where knowledge is gradually encoded into the model parameters during training. Research into the nature of ICL is extensive, ranging from mechanistic accounts including induction heads (Olsson et al., 2022) and function vectors (Todd et al., 2023), to theoretical frameworks that interpret ICL as a form of implicit optimization or Bayesian inference (Von Oswald et al., 2023; Dai et al., 2022; Xie et al., 2021). Most relevant to this work, however, is the finding that the emergence of ICL relies on specific statistical properties of the training data (Chan et al., 2022).

While the emergence of ICL can be productively understood in terms of rational adaptation to environmental pressures (Wurgaft et al., 2025), this perspective has limitations. By focusing on what the asymptotically optimal strategy is (i.e., ICL or IWL), it struggles to account for the learning dynamics, particularly the factors that precipitate transitions between strategies. This gap is significant because empirical studies show that ICL can be transient, emerging early in training only to diminish or be superseded by IWL as the model converges (Singh et al., 2023).

In this paper, we argue that evolutionary theory offers a powerful explanatory lens for understanding these learning dynamics. Specifically, we examine the environmental factors that drive transitions between two analogous biological strategies: *plasticity* and *evolution*. Plasticity is the capacity of

a single genotype to produce different phenotypes (e.g., morphological or behavioral changes) in response to environmental cues. It allows for adaptation within an individual's lifetime, mirroring how ICL modulates inference based on a prompt. Conversely, evolution involves the slow modification of the genotype across generations via selection to produce specific adaptive traits, analogous to updating model parameters via IWL. Furthermore, just as evolution gives rise to plasticity, the capacity for ICL is acquired through the process of IWL.

Evolutionary theory suggests that the reliability of information across timescales determines which adaptive strategy is favored (Stearns, 1989). Plasticity is generally selected for in fluctuating environments, provided reliable cues are available to guide the adaptive response (Stearns, 1989; Stephens, 1989). However, plasticity tends to be selected against when the environment is stable enough to obviate the need for flexibility, when reliable cues are unavailable, or when the inherent costs of plasticity are prohibitive (Stearns, 1989; Stephens, 1989; DeWitt et al., 1998). Furthermore, *genetic assimilation* describes the evolutionary process by which plastic traits become fixed, eventually rendering their expression insensitive to the cues that originally produced them (Waddington, 1953; Pigliucci et al., 2006). This phenomenon parallels the empirically observed transience of ICL, where the model's reliance on the prompt diminishes as the task is consolidated into weights (Singh et al., 2023).

We hypothesize that analogous factors govern the competition between ICL and IWL in Transformers, determining which strategy dominates at different stages of training. To test this, we operationalize two key dimensions: *cue reliability*, defined as the sufficiency of information within a single prompt to specify the target task, and *environmental stability*, defined as the invariance of the target task across different prompts encountered during training. We manipulate these dimensions in two controlled settings: parametric sinusoid regression and few-shot Omniglot classification. Our results confirm that the evolutionary principle regarding environmental predictability across timescales accurately predicts the learning dynamics of Transformers. This suggests that an ecological view of training environments can offer principled guidance for developing new training methodologies.

## 2 ENVIRONMENTAL PREDICTABILITY

Our general approach relies on drawing a correspondence between aspects of biological adaptation and the learning dynamics observed in Transformers. Fundamentally, we view both systems as optimizing for environmental predictability: the strategy selected depends on the timescale at which information is most reliable. To make these parallels concrete, we consider two examples that illustrate the specific phenomena we investigate.

**Plasticity.** The classic example of plasticity is the defense adaptation of the water flea, Daphnia (Tollrian & Harvell, 1999). When Daphnia detect chemical cues (kairomones) released by predators, they develop a protective helmet and tail spine. In the absence of these cues, individuals with the exact same genotype develop a standard, non-defensive morphology to conserve energy. Just as the Daphnia phenotype is determined by the interaction of a fixed genotype with an environmental cue, a Transformer's output is determined by the interaction of fixed weights with the context provided in the prompt. In both cases, the system adapts its behavior to the current environment without modifying its underlying genotype or weights.

**Genetic assimilation.** The phenomenon where ICL emerges early in training but is later superseded by IWL parallels genetic assimilation, a concept famously demonstrated by Waddington's selection experiments on Drosophila (Waddington, 1953). In these experiments, Waddington exposed fruit fly pupae to a heat shock (an environmental cue), which caused a fraction of them to develop a cross-veinless wing structure (a plastic response). By selectively breeding only those individuals that exhibited the trait, he eventually produced a lineage where the cross-veinless trait appeared without the heat shock. Similarly, a Transformer initially reliant on prompts (ICL) will, given stable training conditions, consolidate the capability into its weights (IWL), rendering the inference insensitive to the contextual cues that originally guided it (Singh et al., 2023).

At its core, these two phenomena hinge on distinct aspects of environmental predictability: the reliability of cues to guide adaptation within a lifetime, and the stability of the environment to facilitate evolution across generations. Both dimensions ultimately derive from the predictive power of information at different timescales — specifically, how accurately experience (whether gathered

within a single lifetime or accumulated across generations) anticipates optimal behavior. If this correspondence between biological adaptation and learning in Transformers holds, then systematically manipulating predictability at different timescales (e.g., via noise) should allow us to determine the preferred learning strategy in Transformers. The following section details how we operationalize this experimentally in two controlled training environments.

## 3 EXPERIMENTAL SETUP

ICL is widely viewed as an emergent form of meta-learning. Accordingly, we use two standard benchmarks from the meta-learning literature: parametric sinusoid regression (Finn et al., 2017) and few-shot Omniglot classification (Lake et al., 2015). These environments allow us to systematically manipulate our proposed dimensions of environmental predictability (cue reliability and environmental stability) across both continuous and discrete domains.

### 3.1 SINUSOID REGRESSION

In the sinusoid regression task (Finn et al., 2017), the objective is to predict the output of a target function $f_t(x) = A_t \sin(x + \phi_t)$ (see Figure 1). Each training episode consists of a prompt of $N = 10$ examples, $\{(x_i, y_i)\}_{i=1}^{N}$, followed by a query input $x_q$ for which the model must predict the ground truth $y_q = f_t(x_q)$. The inputs $x$ are sampled uniformly from $[-\pi, \pi]$.

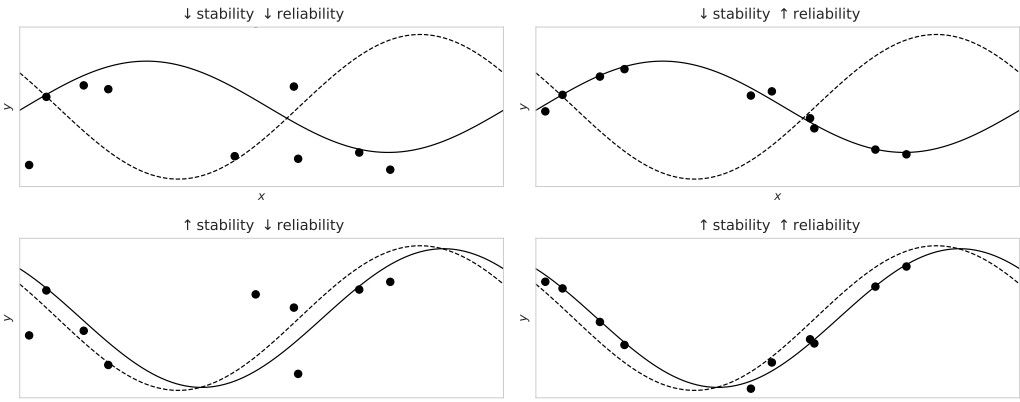

Figure 1: Sinusoid regression under varying predictability conditions. Each panel shows prompt examples (black dots) for a given target function at timestep $t$ (solid line), and the function at the preceding timestep $t - 1$ (dashed line). Top row depicts volatile environments, meaning the target function changes significantly at each step. Bottom row depicts stable environments, where the target function is more stable across time. Left column depicts unreliable cues, where prompt examples are noisy. Right column depicts reliable cues, where prompt examples are less noisy and closer to the target function.

**Cue reliability.** We manipulate this by adjusting the variance, $\sigma^2$, of the Gaussian observation noise $\delta_i \sim \mathcal{N}(0, \sigma^2)$ added to the prompt targets $y_i$. A lower $\sigma^2$ corresponds to higher cue reliability, as the prompt examples more faithfully reflect the true underlying function $f_t$.

**Environmental stability.** We manipulate this by controlling the parameter $\alpha$ in an AR(1) process that governs the evolution of the task parameters $\theta_t = [A_t, \phi_t]^\top$ across training steps $t$. The parameter $\alpha \in [0, 1]$ dictates the correlation between successive tasks, where $\alpha \approx 1$ indicates high stability. The update rule is given by:

$$\theta_t = \alpha \theta_{t-1} + (1 - \alpha)\tilde{\theta}_t \tag{1}$$

where $\tilde{\theta}_t$ represents random innovations drawn from the base task priors: amplitude $A \sim \mathcal{U}[0.5, 1.5]$ and phase $\phi \sim \mathcal{U}[0, 2\pi]$. This formulation ensures mean-reversion towards the prior distribution, allowing us to vary stability without drifting outside the solvable task distribution.

## 3.2 OMNIGLOT CLASSIFICATION

We adapt the few-shot Omniglot benchmark (Lake et al., 2015) to construct a dynamic binary classification task (see Figure 2). At each training step $t$, a global mapping $M_t : \mathcal{C} \to \{0, 1\}$ assigns a hidden binary label to every character class $c$ in the Omniglot vocabulary $\mathcal{C}$ ($|\mathcal{C}| = 1623$). We define the target function $f_t(x) = M_t(c_x)$, where $c_x$ is the character class of image $x$.

Each episode consists of a prompt with $N = 2$ image-label pairs $\{(x_i, y_i)\}_{i=1}^N$ and a query image $x_q$. The query image belongs to character class $c_q$, and the model must predict its current true label $M_t(c_q)$. Crucially, we construct the prompt such that one example matches the query's class ($c_k = c_q$). This setup explicitly creates a competition between strategies: the model can solve the task via ICL (copying the label $y_k$ from the matching prompt example) or via IWL (relying on the mapping $M_t(c_q)$ stored in its weights).

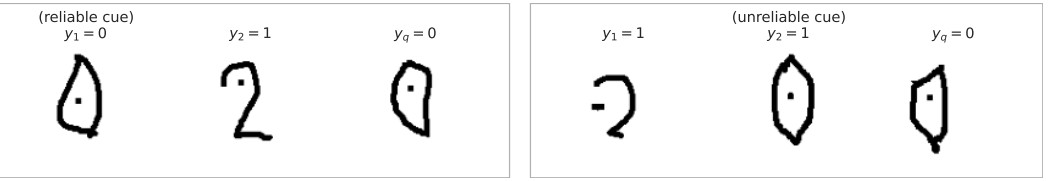

Figure 2: Omniglot classification episodes. The prompt on the left contains a reliable cue (prompt label $y_1$ for character $c_1 = c_q$ matches $M_t(c_q)$). The prompt on the right contains an unreliable cue (prompt label $y_2$ for $c_2 = c_q$ is flipped relative to $M_t(c_q)$).

**Cue reliability.** We operationalize reliability by controlling the probability, $\rho$, that prompt labels are accurate. For each example $i$ in the prompt, the provided label $y_i$ corresponds to the true mapping $M_t(c_i)$ with probability $\rho$, and is flipped otherwise:

$$y_i = \begin{cases} M_t(c_i) & \text{with probability } \rho \\ 1 - M_t(c_i) & \text{with probability } 1 - \rho \end{cases} \tag{2}$$

**Environmental stability.** We operationalize stability by controlling the persistence of the global mapping $M_t$ over time. The initial mapping $M_0(c)$ is sampled from a Bernoulli(0.5) distribution. For subsequent steps $t > 0$, each character's label persists with probability $\gamma$:

$$M_t(c) = \begin{cases} M_{t-1}(c) & \text{with probability } \gamma \\ 1 - M_{t-1}(c) & \text{with probability } 1 - \gamma \end{cases} \tag{3}$$

A higher $\gamma$ indicates greater stability. This process induces a first-order temporal autocorrelation of $\alpha = 2\gamma - 1$ for each character's label trajectory.

## 3.3 MODEL AND TRAINING

All experiments use a decoder-only Transformer with 4 layers, 4 attention heads per layer, an embedding dimension of 128, and learned positional encodings. Input processing varies by task. For the sinusoid regression task, scalar inputs and outputs are linearly projected to the embedding dimension. For Omniglot classification, character images are processed by a shallow ResNet with three stages, each with 2 residual blocks, and outputting 16, 32, and 64 channels per stage, respectively. This ResNet is trained jointly with the Transformer.

Models are optimized to minimize either the mean squared error (MSE) for sinusoid regression or the binary cross-entropy (BCE) for Omniglot classification, calculated on their predictions for the query item ($y_q$). We use the AdamW optimizer (Loshchilov & Hutter, 2017) with a peak learning rate of $1 \times 10^{-4}$, subject to a cosine decay schedule following 1,000 warmup steps. All models are trained for a total of 50,000 training steps, using a batch size of 128.

### 3.4 EVALUATION

To quantify the model's preference for ICL versus IWL, we use an evaluation protocol inspired by Chan et al. (2022). This involves constructing evaluation prompts where the underlying target task explicitly conflicts with the current training environment.

**Evaluation prompts.** We generate these prompts by sampling an evaluation task, $f_e$, from the prior distribution such that it is independent of the current training task, $f_t$. We then construct a prompt $\{(x_i, y_i)\}_{i=1}^N$ using labels derived from this evaluation task ($y_i = f_e(x_i)$).

**Target definitions.** For a given query $x_q$, this setup defines two conflicting prediction targets:

- The ICL target, $y_{\text{ICL}} = f_e(x_q)$, corresponds to the label implied by the prompt; a model predicting this value is relying on context (ICL).

- The IWL target, $y_{\text{IWL}} = f_t(x_q)$, corresponds to the label implied by the current training environment; a model predicting this value disregards the prompt in favor of its weight-based encoding of the environment (IWL).

**Quantifying strategy preference.** We quantify the model's strategy preference by computing the prediction error relative to both targets ($E_{\text{ICL}}$ and $E_{\text{IWL}}$) using the task-appropriate metric (MSE for Sinusoid, BCE for Omniglot). We define the ICL preference score as:

$$S_{\text{ICL}} = \frac{E_{\text{IWL}}}{E_{\text{ICL}} + E_{\text{IWL}}} \tag{4}$$

An $S_{\text{ICL}} \approx 1$ indicates pure reliance on context, while $S_{\text{ICL}} \approx 0$ indicates pure reliance on weights.

## 4 RESULTS

Our experiments reveal that the learning strategies adopted by Transformers are systematically shaped by the predictability of their training environment, in ways that closely parallel adaptive mechanisms in evolutionary biology. We first examine how environmental stability and cue reliability governs the asymptotic preference for ICL versus IWL. We then explore the learning dynamics between these strategies, revealing task-dependent patterns of transience (ICL → IWL and vice versa). Finally, we show that the cost of acquiring ICL versus IWL governs these dynamics. Together, these results reconcile a diverse set of learning phenomena in Transformers with principles of biological adaptation.

### 4.1 DETERMINANTS OF ASYMPTOTIC STRATEGY

Figure 3 shows how dimensions of environmental predictability influence the model's asymptotic strategy ($t = 50,000$), supporting our core hypothesis that information reliability across timescales governs the trade-off between ICL and IWL.

**Stable environments favor ICL.** In sinusoid regression (Figure 3, top), increasing environmental stability ($\uparrow \alpha$) leads to a precipitous decline in ICL preference. As $\alpha \to 1$, the model shifts almost entirely to IWL. This recapitulates the biological principle of genetic assimilation: when environmental features are stable across generations, selection favors the permanent encoding of traits over incurring cost of plasticity (Stearns, 1989; DeWitt et al., 1998). Similarly, in the Omniglot task (Figure 3, bottom), ICL preference plummets as $\alpha \to 1$. This reflects a transition where the global mapping becomes invariant, allowing the model to consolidate the task into its weights and render the prompt redundant.

**Volatile environments with reliable cues favor ICL.** Conversely, when environmental stability is low, the model relies heavily on ICL, provided that reliable cues are available. In sinusoid regression, reliable cues ($\downarrow \sigma^2$, lighter lines) fosters strong ICL preference. This aligns with the evolutionary view that plasticity is favored specifically when environmental fluctuations render fixed traits maladaptive, but only if accurate cues exist to guide the plastic response (Stephens, 1989; Scheiner, 1993). In Omniglot, this effect is even more pronounced: across a broad range of instability ($\alpha < 0.95$), cue reliability is the primary determinant of strategy.

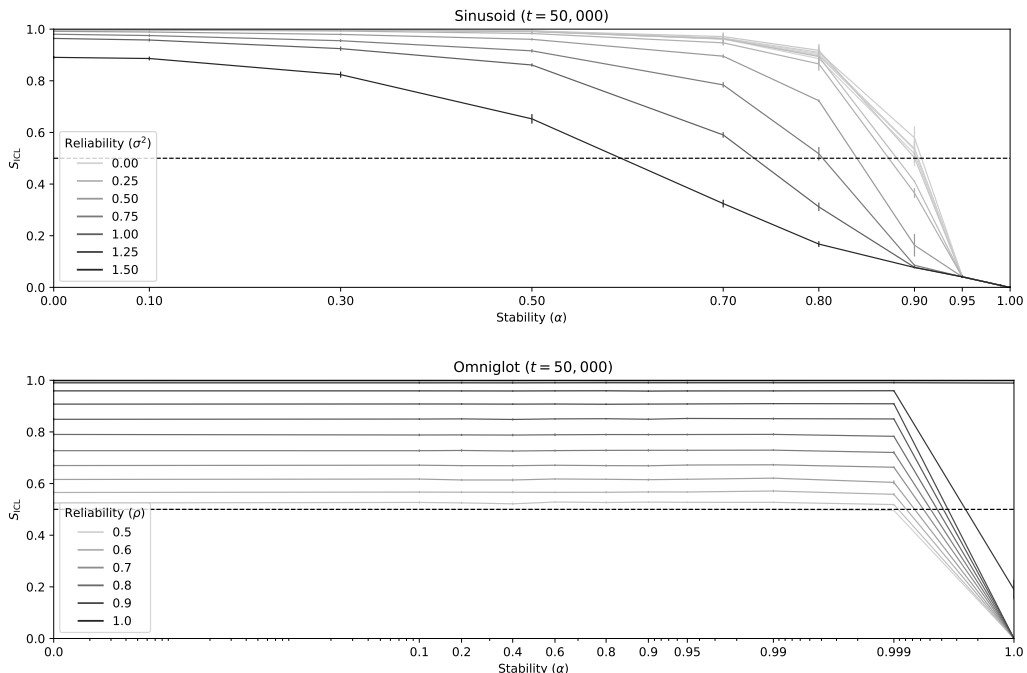

Figure 3: The asymptotic strategy of converged models ($t = 50,000$). The ICL preference score ($S_{\text{ICL}}$) is plotted as a function of environmental stability (x-axis) and cue reliability (line brightness). The dashed line at $S_{\text{ICL}} = 0.5$ marks the crossover point. In sinusoid regression (top), stability is parameterized by the AR(1) coefficient $\alpha$. Cue reliability increases with decreasing noise variance $\sigma^2$ (lighter lines). In Omniglot classification (bottom), stability is parameterized by the mapping persistence $\alpha$ (logit scale). Cue reliability increases with label correctness probability $\rho$ (lighter lines).

**When strategies are equally effective.** Interestingly, we observe a deviation from biology in the Omniglot task. With perfectly reliable cues ($\rho = 1$), the model maintains a strong preference for ICL even when the environment is perfectly stable (Figure 3, bottom right). Evolutionary theory suggests that plasticity is selected against in stable environments due to maintenance costs (DeWitt et al., 1998). The persistence of ICL here implies that, for Transformers, the maintenance cost of ICL may be negligible — or, crucially, that the acquisition cost of the alternative strategy (IWL) is prohibitively high. This suggests that when both strategies are asymptotically effective, the model's preference is driven by the difficulty of learning each solution. We formalize this intuition in Section 4.3.

### 4.2 TASK-DEPENDENT PATTERNS OF TRANSIENCE

Beyond asymptotic strategies, the learning trajectories reveal shifts in strategy preference during training — a phenomenon known as *transience* (Singh et al., 2023; Panwar et al., 2023; Singh et al., 2025). As shown in Figure 4, the direction of change between ICL and IWL is highly task-dependent.

**ICL transience (ICL → IWL).** In the Omniglot task under high environmental stability (Figure 4, top), we observe the phenomenon of ICL transience (Singh et al., 2023). The model exhibits a strong preference for ICL early in training. However, as training progresses, the model gradually shifts towards an IWL solution. This progression parallels genetic assimilation, where a trait initially induced by environmental cues (plasticity) becomes genetically encoded under prolonged, stable selection, eventually rendering the cue redundant (Waddington, 1953; Pigliucci et al., 2006).

**IWL transience (IWL → ICL).** In contrast, the sinusoid regression task (Figure 4, bottom) exhibits the opposite trajectory: IWL transience (or delayed ICL). Under medium stability ($\alpha = 0.8$), the

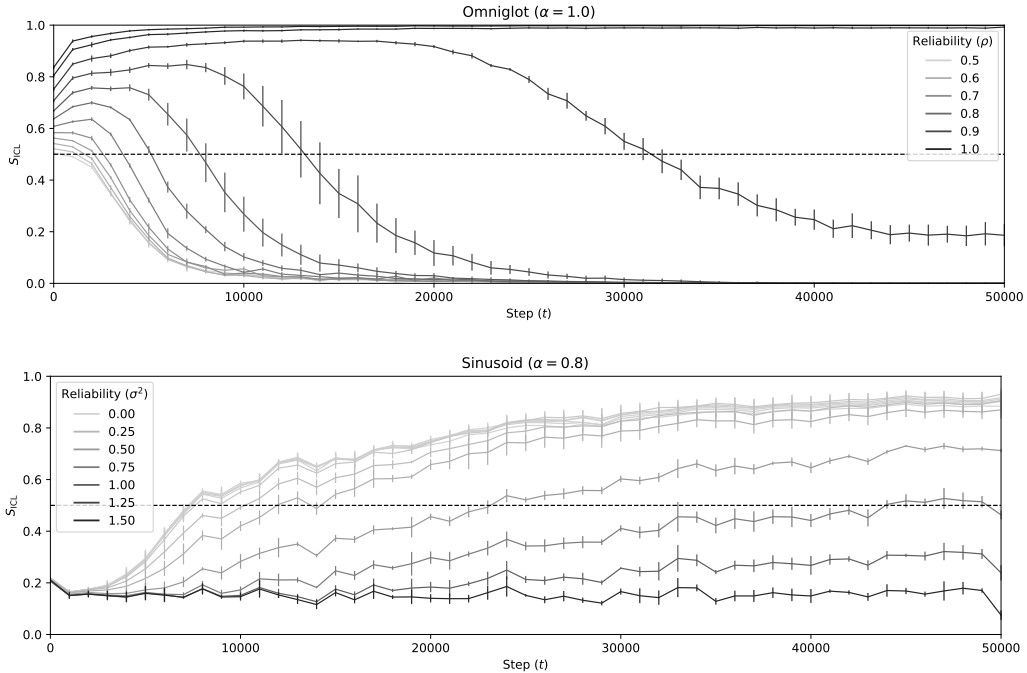

Figure 4: The shift in ICL preference ($S_{\text{ICL}}$) over training steps ($t$) reveals opposing directions of transience. Omniglot classification (top) demonstrates ICL transience ($\alpha = 1$). The model initially relies on ICL but gradually consolidates the task into weights. Sinusoid regression (bottom) demonstrates IWL transience ($\alpha = 0.8$). The model initially relies on IWL but gradually improves inferences by using the prompt.

model initially relies on IWL (approximating the function via weights) and exhibits a low preference for context. A strong preference for ICL emerges only later in training, particularly when cues are highly reliable ($\downarrow \sigma^2$). This suggests that for sinusoid regression, an IWL approximation of the target function is easier to learn early on, whereas the circuitry required for ICL develops more slowly. This pattern relates to phenomena like *grokking*, where generalization capability (here, ICL) emerges abruptly after a long period of fitting the training distribution (Power et al., 2022).

### 4.3    COST-BASED ARBITRATION BETWEEN STRATEGIES

The contrasting trajectories observed in Section 4.2 raise the question of why Transformers initially prefer ICL in Omniglot (ICL → IWL), but IWL in Sinusoid (IWL → ICL). These results suggest that learning dynamics are governed not only by the asymptotic optimality of a strategy (which is determined by the environment), but also by the cost of acquiring that strategy. We posit that the strategy that is cheaper to learn (i.e., whose inductive bias closely fits the task structure) will emerge earlier in training, regardless of whether it is the long-term optimal solution. This aligns with the broader literature on *simplicity bias* (Arpit et al., 2017; Goldblum et al., 2023; Deora et al., 2025), while highlighting that simplicity is contingent on the correspondence between the model and the training data.

**Cost of learning for each strategy.** Intuitively, the relative difficulty of IWL and ICL differs across our two tasks. In Omniglot, IWL requires memorizing a global mapping for a large vocabulary ($|\mathcal{C}| = 1623$). This is a high-capacity memorization problem with high sample complexity. In contrast, ICL requires only a local pattern-matching circuit (comparing the query to the prompt) (Olsson et al., 2022; Singh et al., 2024), which is simple for the attention mechanism to discover. Thus, for Omniglot, ICL is cheaper than IWL. In sinusoid regression, the dynamic is reversed. An IWL solution requires learning a single global sine wave approximation, which neural networks can parameterize rapidly. Conversely, ICL requires implementing a regression algorithm within the

forward pass (e.g., gradient descent or ridge regression via attention (Von Oswald et al., 2023)). This is a complex circuit that is difficult to learn. Thus, for sinusoid, IWL is cheaper than ICL.

**Quantifying learning cost.** To formalize this intuition, we adopt the minimum description length (MDL) perspective for quantifying the cost of learning a particular dataset, using the *prequential codelength* (Blier & Ollivier, 2018). Prequential codelength is the cumulative negative log-likelihood (NLL) of each token, computed sequentially by a model strategy (ICL or IWL) trained on all preceding tokens. See Appendix A.3 for a formal description. Intuitively, a strategy that is easier to learn for the particular dataset (more aligned inductive bias) will achieve lower log-likelihoods early in training, resulting in a shorter prequential codelength.[1]

Because our goal is to evaluate the inductive bias of different learning strategies the transformer can adopt (ICL or IWL), rather than the Transformer as a whole, we consider training environments where either ICL or IWL are strongly favored. For ICL, this means a volatile environment ($\alpha = 0.0$) with reliable cues ($\rho = 1.0, \sigma^2 = 0.0$). And for IWL, this means a stable environment ($\alpha = 1.0$) with unreliable cues ($\rho = 0.5, \sigma^2 = 1.5$). We then computed the prequential codelength of each strategy (ICL and IWL) for each task (sinusoid and Omniglot) by keeping a running sum of the NLL (i.e., the loss) during the entire course of training.[2] As shown in Figure 5, the results confirm our intuition. For Omniglot, the cost for ICL is significantly lower than for IWL. Conversely, for sinusoid, the cost for IWL is significantly lower than for ICL. This metric predicts the starting point of the learning trajectories in Figure 4 — the model adopts the cheaper strategy first.

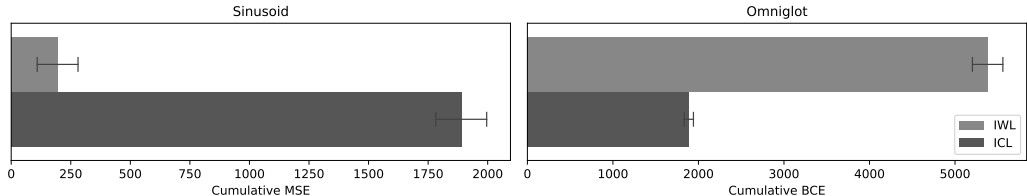

Figure 5: Cost of learning IWL vs ICL, measured as the prequential codelength (cumulative MSE/BCE loss accrued over 50,000 training steps). IWL is less expensive than ICL for Sinusoid, and the reverse is true for Omniglot, showing that the relative cost of IWL vs ICL corresponds to transience dynamics (IWL-first or ICL-first) observed experimentally for the two tasks.

**Manipulating learning cost.** To causally validate that learning cost drives these dynamics, we manipulated the difficulty of the IWL strategy in the Omniglot task. The sample complexity of IWL (memorizing labels for all characters) is governed by the coupon collector problem — to see all $N = |\mathcal{C}|$ characters at least $K$ times requires $\Theta(KN \log N)$ episodes. By reducing the vocabulary size $|\mathcal{C}|$, we reduce the cost of the IWL strategy.

As Figure 6 shows, reducing $|\mathcal{C}|$ from 1623 to 100 drastically changes the learning dynamics. In the standard setting (dark line), the model starts with high ICL preference. However, in the reduced setting (lightest line), the dynamics reverse: the model starts with a preference for IWL ($S_{\text{ICL}} < 0.2$) before eventually switching to ICL. This induced IWL transience (reproducing the dynamic seen in the Sinusoid task) shows that the initial preference of a strategy is determined by its acquisition cost.

Our findings align with recent work linking ICL transience to task structure (Singh et al., 2023) and relative acquisition speeds (Nguyen & Reddy, 2024; Singh et al., 2025; Wurgaft et al., 2025), as well as mechanistic studies on circuit efficiency and competition dynamics (Thilak et al., 2022; Nanda et al., 2023; Varma et al., 2023; Park et al., 2024). Together, they imply that Transformers arbitrate between strategies based on a distinct optimization cost. The framework of *resource rational analysis* (Lieder & Griffiths, 2020), introduced in cognitive science but inspired by earlier work

---

[1]Prequential codelength also exhibits a "catch-up phenomenon" that may be interesting to compare with our transience observations here — complex models have higher cost initially and later require more samples to "catch up", even after they become more optimal (Erven et al., 2012).

[2]It is worth noting that for the Omniglot task, the binary cross-entropy loss is equivalent to the NLL; however for sinusoid regression, the mean squared error loss is monotonically related to the NLL under a Gaussian likelihood.

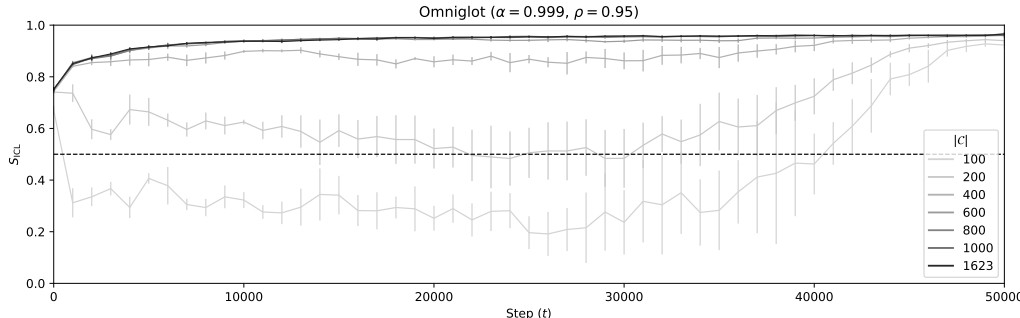

Figure 6: We manipulate the learning cost of the IWL strategy by varying the Omniglot vocabulary size ($|\mathcal{C}|$). We plot ICL preference ($S_{\text{ICL}}$) across training steps ($t$) in a stable environment ($\alpha = 0.999$) with reliable cues ($\rho = 0.95$). In the standard setting ($|\mathcal{C}| = 1623$, dark line), the model exhibits a sustained preference for ICL. Drastically reducing the vocabulary size (e.g., to $|\mathcal{C}| = 100$, lightest line) lowers the cost of IWL, causing the model initially favors IWL before eventually switching to ICL, thereby reversing the pattern of transience previously observed in this task.

in AI (Horvitz, 1987; Russell & Subramanian, 1994), may provide a set of tools for formalizing this tradeoff and offering further insight into the behavior of AI models.

## 5  DISCUSSION

In this paper, we have demonstrated that the competition between ICL and IWL in Transformers is governed by the same principles of environmental predictability that shape biological adaptation. While stability and cue reliability dictate the asymptotic strategy (favoring IWL or ICL, respectively), our analysis of transience reveals that the trajectory of learning is driven by the cost of acquiring each strategy. We find that Transformers consistently adopt the easier strategy first (as determined by their inductive bias for a particular task) while the more costly but asymptotically optimal strategy develops over a longer timescale.

### 5.1  LIMITATIONS AND FUTURE DIRECTIONS

While this work offers foundational insights into the learning dynamics of Transformers, we acknowledge specific limitations that define its scope, as well as promising avenues for future research.

**Task simplification.** Our sinusoid and Omniglot environments are, by design, simplifications of the complex linguistic environments that LLMs are trained in. However, such controlled settings are essential for isolating variables (e.g., environmental predictability) and their causal influence on learning dynamics. Future work should aim to verify these principles in more complex domains.

**Levels of analysis.** Our framework constitutes a computational-level analysis (Marr, 1982; Ku et al., 2025). We characterize the adaptive problem a system faces (optimizing prediction under varying uncertainty) and the functional logic of its solution (ICL vs. IWL). The analogy to evolutionary biology operates at this functional level of adaptation to statistical structure in the environment; we do not claim a direct mechanistic equivalence between IWL and ICL in Transformers and the specific genetic or developmental pathways of biological adaptation.

**The Baldwin effect.** Beyond merely managing uncertainty, a key evolutionary function of plasticity is to accelerate adaptation on rugged fitness landscapes (the *Baldwin Effect*). Plasticity smooths the fitness landscape by enabling individuals to survive in new environments, guiding the population toward regions where genetic assimilation can subsequently fix the trait (Hinton et al., 1987; Fernando et al., 2018). An exciting direction for future research is to determine if an analogous dynamic exists in Transformers — does the early emergence of ICL serve as a scaffold for accelerating the acquisition of IWL solutions for complex tasks?

## 5.2 CONCLUSION

By viewing Transformers through the lens of evolutionary biology, we show that their learning dynamics are governed by the statistical structure of their environments. We find that the preference for ICL versus IWL is strongly determined by dimensions of environmental predictability (specifically, environmental stability and cue reliability). Furthermore, our analysis of learning dynamics reveals that while the environment dictates the asymptotically optimal strategy, the learner's inductive bias leads to transient phases where the model exploits a lower-cost solution first. This framework establishes a more ecological perspective on model behavior, paving the way for training methodologies that cultivate systems capable of robustly and flexibly navigating the diverse and dynamic environments they increasingly encounter.

## ETHICS AND REPRODUCIBILITY STATEMENTS

**Ethics statement.** This work constitutes basic research into the fundamental learning dynamics of Transformers. Our experiments rely exclusively on synthetic data and the publicly available Omniglot benchmark; consequently, the work involves no personally identifiable information and we do not anticipate any direct negative societal impacts.

**Reproducibility statement.** To ensure the reproducibility of our findings, we have provided a detailed description of the model architecture, training protocol, and task generation processes in Section 3. Comprehensive hyperparameter configurations and details regarding compute resources are listed in Appendix A. Furthermore, the complete codebase for data generation, training, and analysis will be made publicly available upon publication.

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

## A    TECHNICAL APPENDICES AND SUPPLEMENTARY MATERIAL

### A.1    PARAMETER SWEEPS AND REPLICATION

To investigate the effects of environmental predictability, we performed dense parameter sweeps for the Sinusoid regression and Omniglot classification tasks. Unless otherwise specified, all results reported in figures are the mean across 3 runs using different random seeds, and plotted error bars represent $\pm 1$ standard error of the mean (SEM).

**Sweep 1: Sinusoid regression**

The following grid of parameter values was used:

- $\alpha \in \{0.0, 0.1, 0.3, 0.5, 0.7, 0.8, 0.9, 0.95, 0.99, 0.999, 1.0\}$
- $\sigma^2 \in \{0.0, 0.005, 0.01, 0.02, 0.05, 0.1, 0.2, 0.3, 0.5, 0.75, 1.0, 1.5\}$

**Sweep 2: Omniglot classification**

The following grid was used, where $\alpha$ is determined from $\gamma$ via $\alpha = 2\gamma - 1$, as described in the main text.

- $\alpha \in \{0., 0.1, 0.2, 0.4, 0.6, 0.8, 0.9, 0.95, 0.99, 0.999, 1.\}$
- $\rho \in \{0.5, 0.55, 0.6, 0.65, 0.7, 0.75, 0.8, 0.85, 0.9, 0.95, 0.99, 1.\}$

**Sweep 3: Omniglot vocabulary size**

To generate Figure 6, environmental stability and cue reliability were held at high values ($\alpha = 0.999$, $\rho = 0.95$), while IWL cost was varied via the vocabulary size:

- $|\mathcal{C}| \in \{100, 200, 400, 600, 800, 1000, 1200, 1400, 1623\}$

These sweeps comprise $(11 \times 12) + (11 \times 12) + 9 = 273$ unique experimental configurations. Including the 3 random seeds per configuration, this resulted in a total of 819 model training runs.

### A.2    COMPUTE RESOURCES

All models were trained on single TPUv3 cores. Each of the 819 training runs took approximately 30 minutes to complete the 50,000 training steps (detailed in Section 3). The total compute usage was approximately 410 TPU-hours.

### A.3    PREQUENTIAL CODELENGTH

The standard and formal definition of prequential codelength is (Blier & Ollivier, 2018):

$$L(D) = -\sum_{t=1}^{n} \log P(y_t | x_t; \theta_{t-1})$$

where $D = ((x_1, y_1), ..., (x_n, y_n))$ is a sequence of observed data points and $\theta_{t-1}$ represents the model parameters trained on the data observed up to time $t - 1$: $((x_1, y_1), ..., (x_{t-1}, y_{t-1}))$.

## B    USE OF LARGE LANGUAGE MODELS

During the preparation of this manuscript, a large language model was used to assist with editing the prose for clarity and to aid in literature discovery. This latter function was particularly valuable for exploring the evolutionary biology literature that provides the conceptual framework for this study. All scientific claims and the final content of the paper were written and verified by the authors.

