# OpenReview forum: "An evolutionary perspective on modes of learning in Transformers"
_ICLR.cc/2026/Conference — ICLR 2026 Poster_

### Official Review · Reviewer_ER3k · 2025-10-27

**Soundness:** 3
**Presentation:** 3
**Contribution:** 3
**Rating:** 6
**Confidence:** 3

**Summary:**

The authors investigate in-context learning versus in-weights learning in Transformers (though I don't see why the experiments couldn't be done with other memory-based architectures, like RNNs).

Adopting meta-learning settings,  both Omniglot and sinusoid regression, they independently manipulate the stability of the (outer-loop, across tasks) task distribution, and the reliability of the (inner-loop, within-task) contextual cues.

Then, testing with a new task allows then to estimate how much the network relies on in-weights or in-context learning, based on whether its answer to the final query input is closer to that predicted from current in-context cues, or from the ongoing training task, respectively.

They show that higher contextual reliability favors in-context learning, while higher task stability favors in-weight learning. Different tasks have different dynamics. Increasing the number of possible classes decreases preference for in-weights learning in the Omniglot task.

**Strengths:**

The question is interesting. The experiments are informative. The review of the literature (on both evolution and ML) is nice.

**Weaknesses:**

There seems to be no fatal flaw in the paper that I can see.

It may be argued that some of the results are not really earth-shattering ("more stability = more overfitting?"), though the additional experiments in Figures 4 and 5 provide more details on the dynamics.

It's not clear how much the "relative cost" hypothesis helps, because there seems to be no precise definition of "cost", except for a-posteriori hardness on IWL? (I note that the parameter used to tune Omniglot task is the total number of classes; however, the sinusoid task has an infinite number of classes, yet it doesn't seem to clearly favor ICL or IWL more than Omniglot, but rather it seems to incline differently for various regimes of stability/reliability).

**Questions:**

If the authors can make their hypotheses a bit more precise and/or actionable it would probably increase the reach of the paper. Other than that I have no pressing questions.

---

> ### Author Response · Authors · 2025-11-23
>
> We thank the reviewer for the positive assessment. We are glad you found the research question interesting and the experiments to be insightful. We especially appreciate your suggestion to quantify the "relative cost hypothesis" – we have done so and added a number of experiments accordingly, which we believe has significantly strengthened the paper. Thank you so much for this suggestion.
>
> _"It may be argued that some of the results are not really earth-shattering ('more stability = more overfitting?')"_
>
> We respectfully clarify a key distinction: IWL is not the same as overfitting, which corresponds to adapting to noise and failing to generalize, and is an orthogonal property of a model's solution. IWL corresponds to learning the underlying function $f_t(x)$ and encoding it into weights of the network, and this can be a solution with good or poor generalization properties. For example, in our experiments in high-stability settings, models that learn IWL solutions still achieve near-zero error on new, unseen query inputs (good generalization and low overfitting).
>
> _"There seems to be no precise definition of 'cost'..."_
>
> Thank you for this great suggestion. We agree that this quantification is an important way to strengthen the paper, and we have now added multiple analyses accordingly. We have now quantified cost in terms of prequential coding cost, which is the cumulative negative log-likelihood of each token, computed sequentially by a model strategy trained on all preceding tokens. This approach aligns with the minimum description length (MDL) perspective for measuring the cost of learning a particular dataset. Intuitively, a strategy that is easier to learn for the particular dataset (more aligned inductive bias) will achieve lower log likelihoods early in training, resulting in a lower total cost. We have performed new experiments using this metric, and we show that it cleanly predicts transience in accordance with the relative-cost hypothesis. In sinusoid regression (where we observe IWL transience), C_IWL < C_ICL. In standard Omniglot (ICL transience), C_ICL < C_IWL.
>
> _"Sinusoid task has an infinite number of classes, yet it doesn't seem to clearly favor ICL... more than Omniglot."_
>
> Thank you for this question. The key is the complexity of the function parameter space, not the output space. Learning the Omniglot mapping requires memorizing 1623 arbitrary bits. Whereas for sinusoid regression, despite infinite outputs, the rule is defined by only 2 parameters (amplitude, phase). The parameters of the sinusoid are cheaper to encode than the omniglot mapping, which is why we see IWL transience for sinusoid, but ICL transience for omniglot. This has been clarified in the second paragraph of Section 4.3.

---

> > ### Comment · Reviewer_ER3k · 2025-11-25
> >
> > The added content is basically incomprehensible and needs *a lot* of explanation.
> >
> > "computed sequentially by a model strategy trained on all preceding tokens." - what is a "model strategy"? Preceding what? How is the training done? How exactly do "ICL" and "IWL" environments differ?
> >
> > Please explain exactly what the quantities in Figure 5 are and how they are computed.

---

> > > ### Author Response · Authors · 2025-11-27
> > >
> > > We are happy to provide more explanation, and we have added this to the appendix as well.
> > >
> > > _'"computed sequentially by a model strategy trained on all preceding tokens." - what is a "model strategy"? Preceding what?'_
> > >
> > > We are here using the term "strategy" in a way consistent with the rest of the paper -- specifically, in-context learning or in-weights learning as strategies for improving inference. We understand that this terminology may not have been obvious. We have updated the text to read "a model strategy (IWL or ICL)."
> > >
> > > The definition here follows a standard formal definition of the prequential codelength (Blier & Ollivier, 2018):
> > >
> > > $$L(D) = -\sum_{t=1}^n \log P(y_t|x_t; \theta_{t-1})$$
> > >
> > > where $D = ((x_1, y_1), ..., (x_n, y_n))$ is a sequence of observed data points and $\theta_{t-1}$ represents the model parameters trained on the data observed up to time $t-1$: $((x_1, y_1), ..., (x_{t-1}, y_{t-1}))$. We have added this definition to the paper as well.
> > >
> > > In other words, the prequential codelength is computed by keeping a running sum of the negative log likelihood (NLL; i.e., the loss) during the entire course of training. Intuitively, this is a good measure of the inductive bias of the model for a particular dataset, because it will achieve lower log-likelihoods early in training, resulting in a smaller cumulative sum.
> > >
> > > _'How is the training done? How exactly do "ICL" and "IWL" environments differ?'_
> > >
> > > Because our goal is to evaluate the inductive bias of different learning strategies the transformer can adopt (ICL or IWL), rather than the Transformer as a whole, we consider training environments where either ICL or IWL are strongly favored. For ICL, this means a volatile environment ($\alpha=0.0$) with reliable cues ($\rho=1.0, \sigma^2=0.0$). And for IWL, this means a stable environment ($\alpha=1.0$) with unreliable cues ($\rho=0.5, \sigma^2=1.5$).
> > >
> > > Training is exactly the same as other experiments, and is described in section 3. Computing the prequential codelength is often called "prequential evaluation," as it is a way of evaluating model-data fit, as opposed to changing the way the model is trained.
> > >
> > > We have added this information to the Appendix, and a pointer to the relevant Appendix.
> > >
> > > _"Please explain exactly what the quantities in Figure 5 are and how they are computed."_
> > >
> > > The prequential codelength is computed by keeping a running sum of the NLL (i.e., the loss) during the entire course of training. It is worth noting that for the Omniglot task, the binary cross-entropy loss is equivalent to the NLL; however for sinusoid regression, the mean squared error loss is monotonically related to the NLL under a Gaussian likelihood.

---

### Official Review · Reviewer_XQNw · 2025-11-01

**Soundness:** 3
**Presentation:** 2
**Contribution:** 3
**Rating:** 4
**Confidence:** 5

**Summary:**

This paper investigates how environmental predictability influences the balance between in-context learning (ICL) and in-weights learning (IWL) in Transformers. The paper draws many analogies to evolutionary biology's phenotypic plasticity and genetic encoding and the circuit learned in transformers. The authors set up two model systems: sinusoidal regression and Omniglot and set up variables representing environmental stability (task consistency across training) and cue reliability (how informative are in-context examples). Key findings include: (1) high stability favors IWL while high cue reliability enhances ICL, (2) learning dynamics show task-dependent transience patterns (both ICL→IWL and IWL→ICL transitions, this latter seems quite novel), and (3) a "relative-cost hypothesis" suggesting that the computational ease of acquiring each strategy determines preference and transition dynamics.

**Strengths:**

S1: The experiments are quite novel and interesting, even though some past studies study curriculum learning where the loss landscape keeps changing, studying this together with the question of ICL vs IWL is genuinely interesting.

S2: The **quality** of the paper's presentation is great, even though the framing seems questionable (see W1). The paper is well-written and figures are clear. The experimental design is clean and easy to understand.

S3: The identification of IWL->ICL transience (in addition to previously documented ICL transience) seems novel and interesting.

**Weaknesses:**

W1: This is my only major concern of the paper. The connection to evolutionary biology seems to be just at the very high level. The framing makes sense, but does not actually offer any falsifiable statements or predictions. While this could have been a nice discussion point, I don't see why this should be a main theme of the paper instead of simply framing it as a study of circuit competition on non stationary losses. Furthermore the training method lacks any kind of evolutionary mechanism such as selection, mutation, reproduction. The authors themselves do acknowledge that there is no direct equivalence. It seems like the paper could be much better linked to other concepts like learning theory, meta-learning frameworks, complexity theory, etc. I really like the experiments and the experiments were genuinely interesting, but I am left confused why such a superficial connection to evolution was made, without discussing core concepts of evolution: G-P mapping, mutations, genetic drift, etc.

However, I am happy to discuss this further, perhaps there is a connection which is genuinely helpful that I'm missing.

W2: Beyond the weaknesses discussed in W1, it seems like the relative-cost hypothesis is a good intuition to have, but at the same time doesn't seem to be too novel compared to classic discussions in simplicity bias, circuit complexity, memorization budget in deep learning.

W3: The choice of learning rate scheduling is questionable when the data distribution is non-stationary. Perhaps this makes the results harder to interpret since it artificially slows down the speed of learning. However, I don't think this will qualitatively change the results too much.

**Questions:**

It would be good to cite some more papers which also explore IWL vs ICL:

https://arxiv.org/abs/2306.04891 <- this paper seems to co-pioneer the findings on transience, although they didn't focus on presenting it that way.
https://arxiv.org/abs/2412.01003 <- seems directly related to the cost of memorizing more processes and also discuss that circuit complexity slows ICL.
https://arxiv.org/abs/2506.17859 <- also seems related to the relative cost hypothesis.
https://arxiv.org/abs/2506.19351 <- discusses an Occam's razor on complexity.

Q1: I'm not so sure if this is possible, but is it possible to decompose the model prediction in the style of https://arxiv.org/abs/2412.01003, i.e. decomposing the probability itself by the ICL vs IWL probabilities?

---

> ### Author Response · Authors · 2025-11-23
>
> We thank the reviewer for the constructive feedback. We are glad you found the experiments to be clean and the identification of IWL -> ICL transience novel and interesting.
>
> _"The connection to evolutionary biology seems to be just at the very high level... instead of simply framing it as a study of circuit competition."_
>
> This is an accurate characterization of our intent – we took inspiration from the evolutionary biology case and it helped us gain insights about what analyses to run to understand these phenomena. We have clarified this relationship in the revised manuscript.
>
> _"Relative-cost hypothesis... doesn't seem to be too novel compared to classic discussions in simplicity bias."_
>
> While simplicity bias has indeed been established in many settings, we believe the novelty of our work lies in applying it to explain the directionality of transience (between ICL and IWL), and its dependence on the parameters of the learning environment (namely, different types of predictability). In fact, we would argue that it is unclear a priori which of the two solutions (ICL- or IWL-based) is the simpler one. Our work documents transience in both directions IWL->ICL and ICL->IWL, and shows that the cost of acquisition (which can indeed be regarded as a type of simplicity bias) can actually favor either direction (IWL first or ICL first). Nonetheless, you raise an important and highly relevant area of research, and we have clarified the connection to this research in Sections 4.3.
>
> _"The choice of learning rate scheduling is questionable when the data distribution is non-stationary."_
>
> We used a linear-warmup cosine-decay learning rate schedule in our experiments as it is standard practice for training Transformers. However, we agree with the reviewer that the non-stationarity of the learning rate could theoretically interact with the non-stationarity of the training distribution. We are currently working to re-run a subset of the experiments with a constant learning rate schedule to verify our findings, and we will update the appendix with these additional results.
>
> _"I'm not so sure if this is possible, but is it possible to decompose the model prediction in the style of https://arxiv.org/abs/2412.01003, i.e. decomposing the probability itself by the ICL vs IWL probabilities?"_
>
> This is an interesting idea! We believe that the S_ICL metric does effectively decompose the prediction, by measuring loss against two distinct targets (y_ICL and y_IWL). However, we believe it is an interesting parallel and we have added discussion of this paper to Section 3.4.
>
> _"It would be good to cite some more papers which also explore IWL vs ICL..."_
>
> We thank you for the citation suggestions, which we have incorporated.

---

> > ### Comment · Reviewer_XQNw · 2025-11-28
> >
> > I thank the authors for their careful reply!
> >
> > ---
> >
> > All replies are clear and the manuscript edits are sound!
> >
> > ---
> >
> > > This is an accurate characterization of our intent – we took inspiration from the evolutionary biology case and it helped us gain insights about what analyses to run to understand these phenomena. We have clarified this relationship in the revised manuscript.
> >
> > My main point slightly remains. I definitively understand and believe the authors' reply that the evolutionary biology point of view genuinely motivated the research.
> >
> > However, I'm not so sure whether such a motivation is worth being projected in the most visible parts of the paper: the title and the abstract. This concern is amplified by the fact that the keyword in discussion is "evolution". Evolution is such a broad concept that many phenomena can be linked to some evolutionary principles, such as lifelong adaptation, nature vs nurture, population genetics, etc. And that being said I'm not so sure whether "a non-stationary training distribution" is enough to frame the whole paper from the evolutionary perspective.
> >
> > Core concepts of evolutionary dynamics such as mutation, selection, G-P mapping, genetic drift, evolvability seems to be missing.
> >
> > My opinion is that the evolutionary perspective could be described densely in the introduction and perhaps briefly in the abstract but the current emphasis is too strong compared to the grounded connection.
> >
> >
> > However, As opinions on title/abstract framing is a relatively subjective one, **I decreased my confidence level by 2.**
> >
> > ---
> >
> > We thank the authors again for their careful replies and happy to discuss more!

---

> > > ### Comment · Reviewer_XQNw · 2025-11-28
> > >
> > > I cannot currently edit the score/confidence, perhaps a temporary openreview bug?
> > >
> > > Meanwhile, still happy to discuss more and for now I will decrease the confidence score when I can.

---

### Official Review · Reviewer_yWAK · 2025-11-01

**Soundness:** 3
**Presentation:** 4
**Contribution:** 3
**Rating:** 6
**Confidence:** 3

**Summary:**

The paper investigates when decoder-only Transformers rely on in-context learning (ICL) versus in-weights learning (IWL). Using an evolutionary analogy (phenotypic plasticity vs genetic encoding), the authors operationalize cue reliability and environmental stability and perform dense parameter sweeps on two controlled tasks (sinusoid regression and Omniglot binary classification). They introduce a preference score (SICL) comparing errors against ICL- and IWL-targets, present asymptotic preference maps and training-time transience dynamics, and posit a qualitative 'relative-cost' hypothesis to explain strategy emergence.

**Strengths:**

The paper comes with a clear, motivating framing that yields testable experimental axes (stability and cue reliability).

The scientific claims are substantiated with systematic empirical evaluation with dense sweeps and temporal analyses across many configurations.

Coherent qualitative findings across two tasks: stability favors IWL; reliable cues favor ICL; transience depends on relative difficulty.

Efficient experimental design allowed broad exploration and reproducibility in principle; hyperparameter grids and compute are reported.

**Weaknesses:**

I might be mistaken but is there a critical and central inconsistency?: SICL is defined as SICL = EIWL / (EICL + EIWL + eps) but interpreted (and plotted) as higher SICL meaning more ICL.

Insufficient robustness: only 3 seeds per configuration; many key effects (thresholds, transience) need more seeds and statistical tests.

Limited external validity: only two simplified tasks and a single small Transformer; applicability to larger models and potentially LLMs or naturalistic domains is untested.

Missing important ablations/controls: prompt-length (N) sweep, encoder-freeze/pretrain ablations for Omniglot, model-capacity sweep, and explicit conflict trials to directly distinguish ICL vs IWL.

Mechanistic evidence is lacking: no attention/head diagnostics, weight-change tracking, probes, or lesioning to support claims of circuit-level implementation or assimilation into weights.

The relative-cost hypothesis is qualitative and unquantified; no direct cost or sample-complexity metrics are provided to predict transience.

**Questions:**

Please correct and clarify the SICL definition and interpretation. If it was a typesetting mistake, state the intended formula and re-run affected figures and analyses. As a sanity check, include results for a synthetic pure-ICL and pure-IWL predictor showing the corrected SICL behaves as intended.

Provide explicit pseudocode for the evaluation protocol: how EICL and EIWL are constructed, how evaluator prompts are sampled, number of evaluation examples per measurement, and how conflict trials are generated and scored.

Increase robustness: re-run key configurations (those showing sharp transitions or notable transience) with >=5-10 seeds and report SEMs/confidence intervals and statistical tests for main claims.

Perform the following ablations/controls: (a) vary prompt length N; (b) freeze and/or pretrain the ResNet encoder for Omniglot to localize effects; (c) sweep model capacity (smaller/larger Transformers); (d) include explicit conflict trials during evaluation and report how often models follow prompt vs internal mapping.

Quantify the relative-cost hypothesis: measure steps-to-target-error for ICL-only and IWL-only baselines, parameter-efficiency, or representational complexity, and test whether these predict observed transience directions.

Add basic mechanistic analyses: track layerwise weight changes over training, analyze attention-head patterns, or use linear probes/lesioning to show distinct circuitry for ICL vs IWL and to support any assimilation claims.

---

> ### Author Response · Authors · 2025-11-23
>
> We thank the reviewer for an excellent and detailed review. We appreciate that you found the work sound, the presentation excellent, and the experimental design efficient.
>
> _"Is there a critical and central inconsistency? ...interpreted (and plotted) as higher SICL meaning more ICL."_
>
> You are absolutely correct regarding the text typo. The equation is correct, as are all plots. However, the text incorrectly stated that "S_ICL > 0.5 indicates IWL preference". We have corrected the text description in the updated version. We thank the reviewer for catching this typo.
>
> _"Insufficient robustness: only 3 seeds... key effects need more seeds."_
>
> We currently report the standard error of the mean (SEM) across runs, which shows tight error bars and statistically significant phase transitions above and below the critical threshold ($S_{ICL} = 0.5$). We note that the seed count was initially limited to accommodate a very dense sweep of the hyperparameter space (273 models per seed). However, we agree that additional seeds are valuable; we are working to run these additional seeds and will update the final manuscript with the final error bars.
>
> _"The relative-cost hypothesis is qualitative... no direct cost metrics are provided."_
>
> This is a very good suggestion. We agree that this quantification is an important way to strengthen the paper, and we have now added multiple analyses accordingly. As a small advancement on top of the suggested "steps-to-target-error" metric, we have quantified cost in terms of prequential coding cost, which is the cumulative negative log-likelihood of each token, computed sequentially by a model strategy trained on all preceding tokens. This approach avoids needing to set an arbitrary threshold for target error, and also aligns with the minimum description length (MDL) perspective for measuring the cost of learning a particular dataset. Intuitively, a strategy that is easier to learn for the particular dataset (more aligned inductive bias) will achieve lower log likelihoods early in training, resulting in a lower total cost..
>
> We have performed new experiments using this metric, and we show that it cleanly predicts transience in accordance with the relative-cost hypothesis. In sinusoid regression (where we observe IWL transience), C_IWL < C_ICL. In standard Omniglot (ICL transience), C_ICL < C_IWL. We have also performed causal experiments to manipulate the cost of acquisition by changing the complexity of the task, showing the expected changes in the transience dynamics.
>
> We believe that, together, these experiments strongly strengthen our argument,  and they have been added to the paper in Section 4.3.
>
> _"Perform ablations: prompt-length (N), model capacity, conflict trials."_
>
> Thank you for these suggestions. Prompt length ($N$) is indeed another way of operationalizing cue reliability (the amount of information the prompt contains about the underlying task). We chose noise level as our primary manipulation because increasing $N$ in the Omniglot task introduces nuances: increasing the number of examples of the query class increases informativeness, whereas adding other classes acts as a distractor. However, varying $N$ in the sinusoid task is a clean way of varying cue reliability. We are working to run the sinusoid experiment with varying prompt lengths and will update the appendix with these additional results.
>
> We agree that model capacity is an interesting ablation to run, as it has implications for the learning cost of each strategy. For our main results, we chose a model size sufficiently large to learn both tasks with either strategy perfectly, ensuring we measured preference rather than capability. However, it would be valuable to see how the dynamics shift when the model lacks the capacity for one strategy or the other. We are endeavoring to run these experiments with a capacity-constrained model and will update the appendix with the results.
>
> Regarding conflict trials, we believe that our evaluation protocol is conflict-based. We measure strategy preference ($S_{ICL}$) specifically in scenarios where the information in the context conflicts with the information in the weights (requiring the model to choose which source to prioritize).
>
> _"Mechanistic evidence is lacking..."_
>
> We agree that mechanistic understanding is vital. However, we emphasize the distinct value of a computational-level analysis (in the tradition of Marr) -- by characterizing the adaptive problem the system faces (optimizing prediction under varying uncertainty) our work aims to inform what kinds of mechanisms should be expected to emerge. We have clarified this relationship in Section 5.1. However, to bridge the gap between these levels of analysis, we have added citations to the considerable body of work regarding causal circuits in Omniglot and linear regression.

---

### Author Response · Authors · 2025-11-23
**Common response to reviewers and ACs**

We thank the reviewers for their thoughtful comments and constructive feedback. We appreciate the positive feedback on our  experimental design as "clean" and "efficient" (yWAK, XQNw), and on our findings on learning dynamics as "novel","interesting", "informative", and "substantiated"(yWAK, XQNw, ER3k).

We especially appreciate the suggestion to quantify the relative cost of acquisition for IWL vs ICL strategies (yWAK, ER3k)  – we have undertaken to do so, and run experiments showing that the quantified relative cost does indeed correspond with transience dynamics, as hypothesized. We believe that this addition significantly strengthens the paper, and we are grateful for the suggestion.

We have also clarified Eq. 4 (yWAK): We confirm that our plots are correct (1.0 = ICL preference). However, there was indeed a typo in the text description of Eq. 4 ("SICL > 0.5 indicates IWL preference"), which we have fixed. Thank you for raising this.

Updates are highlighted in red in the newly uploaded manuscript, and more detailed responses for each review are provided below.

---

### Author Response · Authors · 2025-12-03

Dear AC,

To aid in the final decision, we provide a summary of our work and the updates made following the review discussion.

Our paper investigates the factors that drive preference for in-context learning (ICL) and in-weights learning (IWL) in Transformers -- both asymptotically and throughout training. We demonstrate that these dynamics are governed by environmental predictability (specifically stability and cue reliability), mirroring factors that drive preference for analogous adaptation strategies in biology (plasticity and evolution). Furthermore, we show that strategy changes during training are driven by the cost of learning each strategy.

Two reviewers (yWAK, ER3k) recommended acceptance (score 6), highlighting the work's novel perspective and controlled experimental approach. Reviewer XQNw leaned toward rejection (score 4), primarily concerning the framing of the biological connection.

In response to requests from Reviewers yWAK and ER3k, we added a new analysis (Section 4.3, Figure 5) quantifying cost in terms of prequential codelength (cumulative negative log-likelihood accrued during training). This analysis confirms our hypothesis: the strategy with the lower prequential codelength emerges first as a temporary solution while the asymptotically optimal strategy catches up.

Regarding Reviewer XQNw's concern that the biological connection was too high-level, we clarified that evolutionary biology served as the inspiration for our experimental design rather than correspondence at the level of mechanism. Following this clarification, the reviewer acknowledged the validity of our motivation.

We thank the reviewers for their thoughtful comments, which have motivated us to strengthen the theoretical grounding of our work.

Kind regards,
The Authors

---

### Meta-Review · Area_Chair_QqTK · 2026-01-07

**Summary:**

Reviewers generally found the paper interesting, well-motivated, and empirically careful, with a clean experimental design that systematically studies the trade-off between in-context learning (ICL) and in-weights learning (IWL) under controlled notions of environmental predictability. Strengths include dense parameter sweeps, clear empirical patterns, and the identification of bidirectional transience between ICL and IWL.

The main concerns focused on the framing around evolutionary biology, which some reviewers felt was too high-level or overly emphasized relative to the strength of the mechanistic correspondence. Additional concerns included limited robustness (few random seeds), restricted external validity due to simplified tasks and small models, and an initially qualitative formulation of the relative-cost hypothesis.

**Reviewer Concerns:**

The rebuttal addresses several central conceptual concerns by clarifying that the evolutionary perspective is intended as a conceptual inspiration, correcting a technical typo in the ICL preference definition, expanding related work, and quantifying the relative-cost hypothesis via prequential codelength, supported by additional analyses and causal manipulations. These clarifications and additions strengthen the interpretation of the observed transience dynamics and were positively acknowledged by one reviewer.

Other concerns remain only partially addressed. In particular, the reliance on simplified tasks and small models is explicitly acknowledged and discussed as a limitation in the paper, but not resolved empirically through larger-scale or more naturalistic experiments. Questions about robustness, mechanistic understanding, and the appropriate prominence of the evolutionary framing also remain partially open.

**Reviewer Scores:**

The reviewer who initially assigned a score of 4 reduced their confidence following the rebuttal. It is possible, though difficult to assess, that with further discussion their score might also have increased, given that several of their core conceptual concerns were directly addressed. The two reviewers who initially gave borderline-accept scores are less likely to substantially change their evaluations, as the rebuttal primarily strengthens interpretation and analysis rather than introducing fundamentally new empirical evidence.

---

### Decision · Program_Chairs · 2026-01-26

Accept (Poster)